# Effect of the Water Model in Simulations of Protein–Protein Recognition and Association

**DOI:** 10.3390/polym13020176

**Published:** 2021-01-06

**Authors:** Agustí Emperador, Ramon Crehuet, Elvira Guàrdia

**Affiliations:** 1Department of Physics, Universitat Politècnica de Catalunya, B4-B5 Campus Nord, Jordi Girona 1-3, 08034 Barcelona, Spain; elvira.guardia@upc.edu; 2CSIC-Institute for Advanced Chemistry of Catalonia (IQAC), Jordi Girona 18-26, 08034 Barcelona, Spain; ramon.crehuet@iqac.csic.es

**Keywords:** water model, protein association, protein–protein interaction, ubiquitin, ACTR, molecular dynamics

## Abstract

We study self-association of ubiquitin and the disordered protein ACTR using the most commonly used water models. We find that dissociation events are found only with TIP4P-EW and TIP4P/2005, while the widely used TIP3P water model produces straightforward aggregation of the molecules due to the absence of dissociation events. We also find that TIP4P/2005 is the only water model that reproduces the fast association/dissociation dynamics of ubiquitin and best identifies its binding surface. Our results show the critical role of the water model in the description of protein–protein interactions and binding.

## 1. Introduction

The achievement of unprecedented time scales in molecular dynamics (MD) simulations thanks to the increase in computational power has unveiled inaccuracies in the force fields and water models used generally so far, deficiencies which remained unnoticed in older simulations below the μs time scale.

A serious limitation of current molecular models is the inability to predict the correct binding of proteins in molecular dynamics simulations. Previous works have shown that current water models and force fields tend to produce binding configurations in clear disagreement with experimentally observed interfaces [1], leading to spurious formation of aggregates in simulations at concentrations where the proteins are known to remain soluble [2,3]. This spurious tendency to protein association has been related to the general tendency to produce overly collapsed structural ensembles of disordered proteins [4].

In this work we compare the results of the simulation of protein solutions using four of the most commonly used water models: the widely used TIP3P [5] water model, the SPC/E [6] model (an improved version of the older SPC model) and the newer TIP4P-EW [7] (the improved version of TIP4P) and TIP4P/2005 [8] models. We focus on the association/dissociation balance obtained with each water model, and the accuracy to reproduce the experimental binding interface in the case of ubiquitin, a very stable small protein. We have also used, as test system a small disordered protein, the activator for thyroid hormone and retinoid receptor (ACTR). These two proteins are extreme cases regarding protein flexibility and stability.

## 2. Methods

We simulated a system composed of two ubiquitin molecules in a simulation box of a size of 80 Å with periodic boundary conditions, corresponding to a solution with a concentration of 6.5 mM, using different water models. We have used the highly accurate Amber99SB-ILDN [9] force field for the simulations with TIP3P, SPC/E and TIP4P-EW water, while for the simulation with TIP4P/2005 water we have used the Amber03w [10] force field. The parameters and main physical properties of these water models can be found in Table 1 of [11].

All the simulations started from the same system configuration, two ubiquitin molecules (PDB ID code 1UBQ) far apart. Simulations were made in the NVT ensemble. Particle-Mesh Ewald summations were used to treat electrostatics. The cut-off of nonbinding interactions was at 10 Å and the MD time step was 2 fs.

For the simulations of ACTR we placed two molecules in a simulation box of a size of 69 Å, corresponding to a concentration of 10 mM. The starting conformations of the disordered ACTR were generated randomly and energy minimized. We chose an initial system configuration such that the two ACTR molecules were in an extended conformation and far apart. The other details of the simulation were the same as for the simulations of ubiquitin.

## 3. Results

NMR experiments show that ubiquitin forms transient low-affinity noncovalent dimers defined by a large interface where many relative orientations are possible [1]. The binding interface is the beta-sheet surface of ubiquitin, formed by the residues 4–12, 42–51 and 62–71.

We show in Figure 1 the minimum distance between the two ubiquitin molecules during the MD trajectories with the four water models. We find reversible binding only in the simulation with TIP4P/2005 water and Amber03w force field, while with the water models used with the Amber99SB-ILDN force field the system eventually gets trapped forming bound structures which are in disagreement with the experimental interface. This collapse is straightforward for TIP3P and SPC/E water, but in the case of the TIP4P-EW water model several dissociations occur before the molecules get trapped in a bound configuration after 250 ns in the MD trajectory.

We show in Figure 2 the distribution of the intermolecular contacts formed during the simulations along the sequence of the ubiquitin molecule. It can be observed that in the simulation with TIP3P and SPC/E the number of contacts found in the experimental binding interface is negligible, while many contacts are found there in the simulation with TIP4P-EW and even more with TIP4P/2005.

Despite abundant contacts being made by residues in the experimental interface for the four-point water models TIP4P-EW and TIP4P/2005, it can be observed in the contact maps of Figure 3 that almost all of them are contacts with residues outside the experimental interface, therefore the simulations do not fully agree with the experimental observations on the noncovalent binding of ubiquitin.

A very recent study by Li and Buck [12] of a 2 μs long simulation with TIP3P water and the very new CHARMM36m force field [13] of an ubiquitin solution at 4.8 mM concentration, known from experiment to be the concentration where 50% of the solute remains unbound, found that the two molecules were in contact only for 34% of time along the trajectory, but the binding interfaces that they found neither coincided with the experimental interface.

With the purpose of assessing quantitatively the difference with the real association propensity of ubiquitin we made a simulation with TIP4P/2005 water of the solution at 5 mM. Despite our longer simulation (see Figure 4), the long dissociation times reduce the statistics of dissociation events, hampering a direct comparison of the fraction of dimer configurations sampled along our MD simulation and the experimental dimer fraction. Such a comparison would demand an amount of MD trajectories far beyond the scope of this work. We find the two ubiquitin molecules in contact for 65% of time in our MD simulation trajectory. We also show in Figure 4 the number of contacts between the two molecules along the trajectory, and in green the number of contacts between residues of the experimental interface of both molecules, which are found to be much lower that the total number of intermolecular contacts. We also show in Figure 4 the RMSD in respect to the starting structure (the crystallographic structure of ubiquitin) for both molecules.

Despite the fact that simulations with TIP4P/2005 water show a higher RMSD respect to the crystal structure of ubiquitin than the ms time scale simulation with TIP3P of Piana et al. [14], showing more flexibility, we do not observe any clear correlation between the formation of intermolecular contacts and changes in the RMSD of each molecule along the trajectory, showing the stability of ubiquitin. The stability of this molecule is more evident in Figure 5, where we show the 2D-RMSD of each molecule along the trajectory. It can be observed that the average RMSD between two structures of the molecule in distant moments of the trajectory is around 2 Å.

We now focus on the stability of the bound structures found in the simulations with different water models. We show in Figure 6 several snapshots of bound structures found in the simulations with TIP3P, TIP4P-EW and TIP4P/2005 water. It can be observed that in all cases the experimental binding interface of one of the molecules, which we have used to superimpose all the snapshots, binds with a surface of the other molecule different than the experimental interface. In all cases the snapshots shown cover an interval of 100 ns in the MD trajectory. Noticeably in the simulation with TIP4P/2005 water the second molecule rotates and slides over the binding surface, while with TIP3P and also TIP4P-EW the second protein stays in a very stable position relative to the first one. In the case of the TIP4P/2005 simulation, instead of getting trapped in a well defined binding conformation, the second molecule rotates and slides sampling different contact configurations with the reference molecule, in agreement with the observations of NMR experiments [1], until it dissociates. We consider that the weaker protein-water affinity [11] in the TIP3P and TIP4P-EW water models combined with the Amber99SB-ILDN force field overstabilizes these nonspecific protein–protein interactions in binding interfaces different from the experimental one, trapping the system in a bound structure and preventing dissociation.

Abriata and Dal Peraro [2] simulated a system of three ubiquitin molecules with a simulation box of a size corresponding to the same concentration of 5 mM, using the TIP3P water model. In the majority of cases they found overall collapse of the proteins in the simulation, finding only one dissociation event in 10 independent simulations nearly 1 μs long (all the simulations starting with the three molecules apart). We have made a 5 μs long simulation of the same system using the TIP4P/2005 water model, with a starting configuration equivalent to those used in [2], and we have found several dissociations along the simulation (see Figure 7), in agreement with the results obtained in the simulations of two molecules.

We also show in Figure 7 the number of intermolecular contacts formed by each molecule along the trajectory: blue line for molecule A, red for B and green for C. We find that molecule A (blue) gets dissociated from the other two molecules several times along the trajectory. We also find that the most usual configuration of the system is one trimer with a linear topology ABC or ACB, which frequently dissociates by detaching molecule A or molecule C. In a minority of cases a compact but unstable aggregate with triangular topology is found. Overall our results coincide with those observed in the simulations with TIP3P water by Abriata and Dal Peraro [2] in the predominance of the linear topology trimer, but with the relevant difference that with TIP4P/2005 water we find many dissociation events where one molecule detaches from the trimer.

Our findings for ubiquitin seem to indicate that the statibility of intermolecular contacts depends fundamentally on the interaction of water with protein atoms, implemented in the water model and force field [11], rather than on global characteristics of the protein, like its order or stability. To confirm this hypothesis we made simulations of the disordered protein ACTR. The activator for thyroid hormone and retinoid receptor (ACTR) is a 46 residue long intrinsically disordered protein with a very low propensity to aggregation due to its very hydrophilic sequence. Best et al. [4] studied the conformational ensemble of this disordered protein, finding that best results where obtained with the TIP4P/2005 water model, although the ensemble was still too collapsed compared to the experimental observations. In the same work they strengthened the water-protein interactions by a 10%, finding a dramatic expansion of the structures sampled by the disordered protein along its dynamics, which allowed to approach better the experimentally observed value of the radius of gyration of the protein.

We made simulations of two ACTR molecules in a box of the size corresponding to a 10 mM concentration. The intermolecular distance in our simulations is shown in Figure 8. It can be observed that, while with the TIP3P and SPC/E models the two molecules rapidly associate and stay bound for the rest of the simulation, with the TIP4P/2005 and TIP4P-EW water models many dissociation events occur. In this case, thanks to the abundance of dissociation events, we have a good statistics of formation and dissociation of dimers. Many collisions occur due to the fact that the unfolded ACTR molecules span a large region of the simulation box. For this protein the results with both four-point water models are equivalent: many binding events occur but the two molecules rapidly dissociate, producing a low 10% of dimers for TIP4P/2005 and an even lower 5% for TIP4P-EW. Therefore both four-point water models seem to reproduce better the association dynamics of ACTR.

We show in Figure 9 the radius of gyration ($R_{g}$) of the ACTR molecules observed in the simulation with each water model (the $R_{g}$ value of both molecules along the MD trajectory has been used to compute the distribution). We observe a strong dependence of $R_{g}$, whose average experimental value is around 25 Å [4], with the water model used, in a similar way as the dissociation frequency does: regarding the three simulations with the Amber99SB-ILDN force field, only the simulation with TIP4P-EW water produces a conformational ensemble where the ACTR molecule is not collapsed, showing some expanded conformations with $R_{g}$ near 20 Å. The best results are produced by the simulation with TIP4P/2005 water, which sample more expanded conformations with $R_{g}>20$Å.

## 4. Conclusions

Our results show that the water model has a critical effect on the protein dissociation rate and consequently on the degree of aggregation found in simulations of protein solutions. We have observed the same tendency in two extreme cases regarding protein stability: the folded, rigid ubiquitin and the disordered, flexible ACTR.

We have found that nonspecific protein–protein interactions in ubiquitin are overstabilized for all four water models, but the stability of these interactions is lower in the case TIP4P/2005 water with the Amber03w force field, which produce a realistic dissociation rate and prevents the formation of aggregates in the simulation. In the case of ACTR with the Amber99SB-ILDN force field changing water model from TIP3P to TIP4P-EW is enough to remove hydrophobic collapse, a trend which can be observed also in the radius of gyration distribution of this disordered protein.

Despite the formation of bound conformations are in partial disagreement with the experimentally observed interface of ubiquitin [1] in the simulations with TIP4P/2005 water, we find that with this water model the fraction of monomers in the solution at the concentration of 5 mM is in qualitative agreement with the experimentally measured 50%. A higher statistics of dissociation events, obtained from more simulations, will allow a more accurate estimation of the fraction of monomers produced with this water model.

We have observed that in all simulations with the Amber99SB-ILDN force field the bound structures of two molecules of ubiquitin stay very stable until dissociation happens (we find dissociation events only for TIP4P-EW water), while in the simulation with Amber03w and TIP4P/2005 water one of the molecules rotates over the other one, sampling different binding conformations, in agreement with the experimental observations [1]. The abundance of nonspecific protein interactions even with TIP4P/2005 water indicates that further improvement of the water models or force fields is necessary to model properly protein-water interaction and reproduce correctly protein binding.

## Figures and Tables

**Figure 1 polymers-13-00176-f001:**
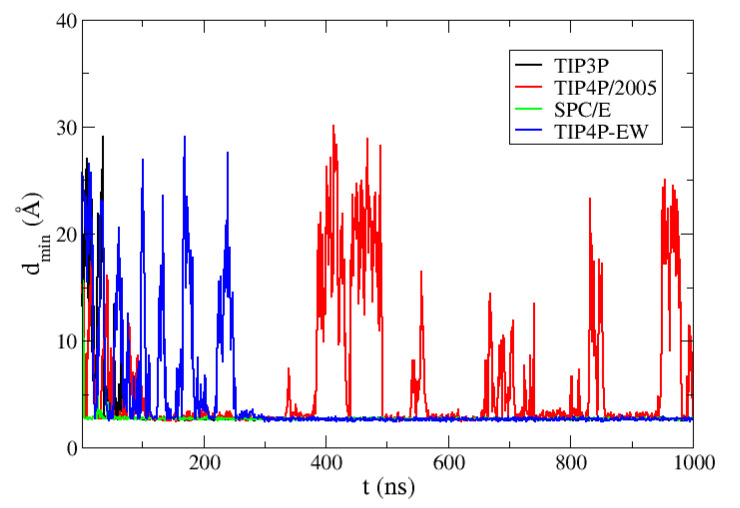
Minimum distance between two ubiquitin molecules in molecular dynamics simulations of a solution of ubiquitin at a concentration of 6.5 mM with four different water models: TIP3P, TIP4P/2005, SPC/E and TIP4P-EW.

**Figure 2 polymers-13-00176-f002:**
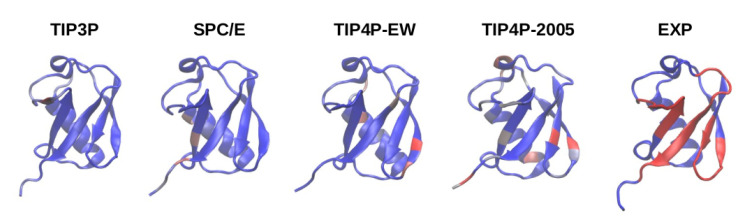
Intermolecular contact frequency along the sequence of the ubiquitin molecule, obtained using TIP3P, TIP4P/2005, SPC/E and TIP4P-EW, using a color scale from blue (zero contacts) to red. At right we show the experimental interface (highlighted in red) for comparison.

**Figure 3 polymers-13-00176-f003:**
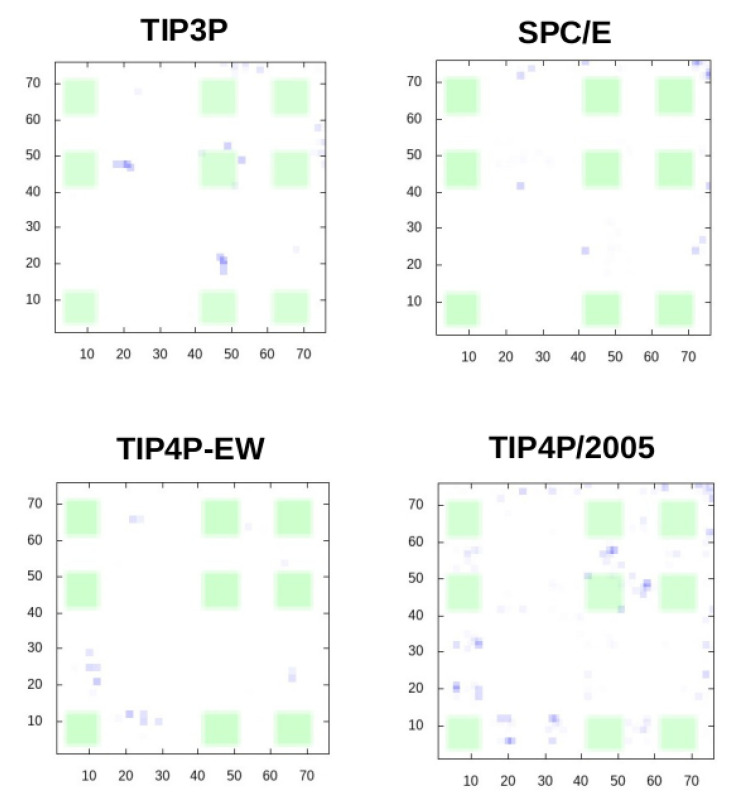
Intermolecular contact maps for the trajectories with the four water models. X,Y axes are the residue number along protein sequence. The regions where contacts are found in the NMR experiments are highlighted in green.

**Figure 4 polymers-13-00176-f004:**
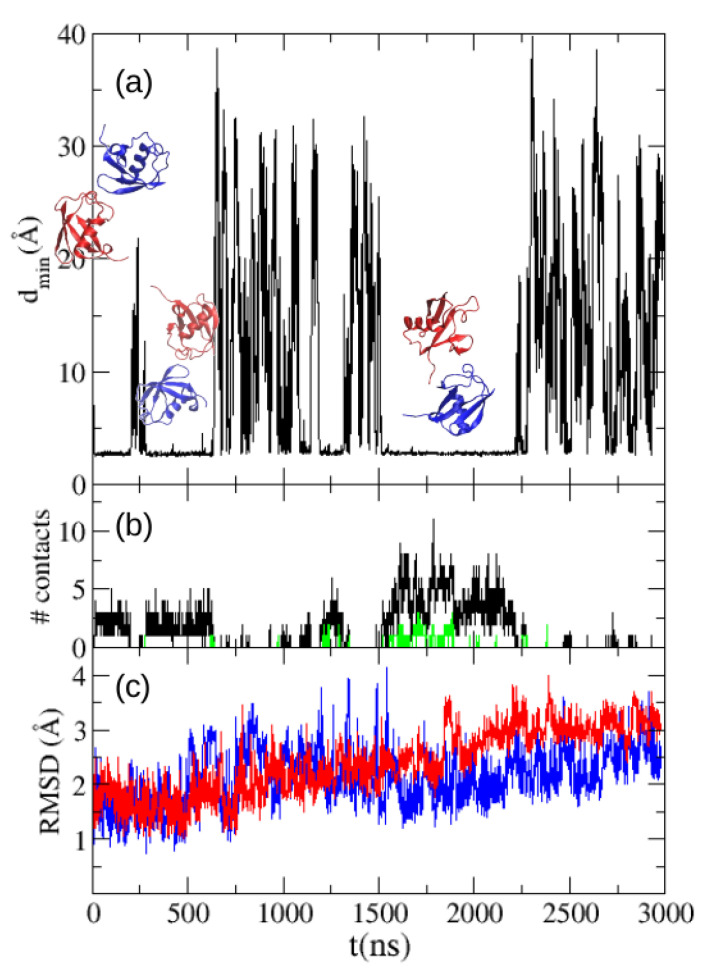
(**a**) Minimum distance between two ubiquitin molecules in a molecular dynamics simulation of a solution of ubiquitin at a concentration of 5 mM with the TIP4P/2005 water model. The most stable bound structures found during the simulations are shown. (**b**) Number of intermolecular contacts found along the simulations. Green line: number of contacts formed between residues of the experimental interface (see main text) (**c**) RMSD to the crystallographic structure of ubiquitin of the two ubiquitin molecules along the simulation.

**Figure 5 polymers-13-00176-f005:**
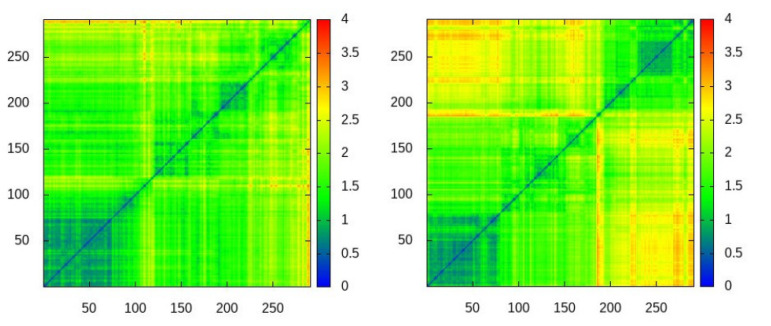
2D-RMSD plot, showing in a color scale the RMSD between all the structures sampled along the MD trajectory, for each of the two ubiquitin molecules in the simulation with TIP4P/2005 water at 5 mM concentration. The scale in both axes is in tens of ns. The same color scale, in Å, is used in both graphs.

**Figure 6 polymers-13-00176-f006:**
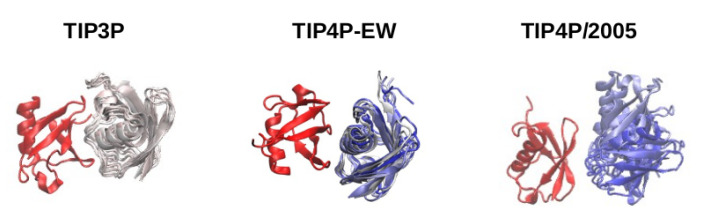
Stability of the bound structures along 100 ns after binding, starting at 100 ns of the trajectory (simulation with TIP3P), 300 ns (TIP4P-EW) and 125 ns (TIP4P/2005). All the snapshots are superimposed to one of the molecules, plotted in red. The snapshots of the other molecule are shown in a time-dependent color scale from white to blue.

**Figure 7 polymers-13-00176-f007:**
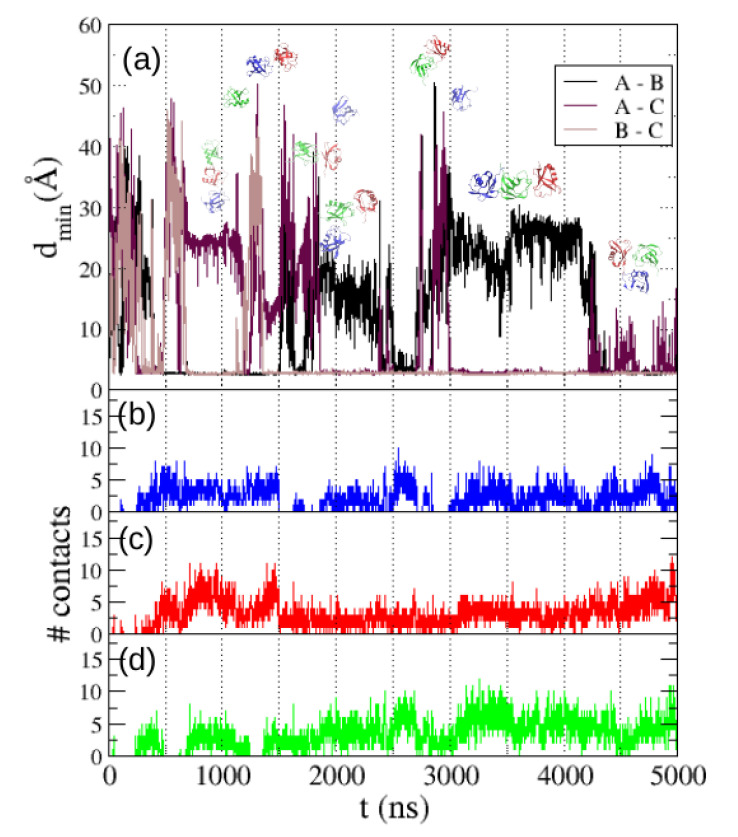
(**a**) Same as Figure 4a when 3 molecules are included in the simulation. The distances between the three molecules (A, B and C) are shown. System configurations at several points of the trajectory are shown. Also shown the number of intermolecular contacts formed by (**b**) molecule A, (**c**) molecule B and (**d**) molecule C.

**Figure 8 polymers-13-00176-f008:**
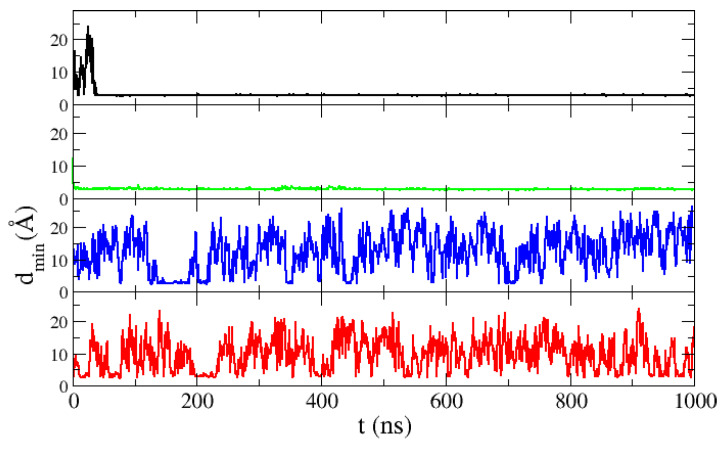
Minimum distance between two ACTR molecules in molecular dynamics simulations of a solution of ACTR at a concentration of 10 mM with the TIP3P (black line), SPC/E (green), TIP4P-EW (blue) and TIP4P/2005 (red) water models.

**Figure 9 polymers-13-00176-f009:**
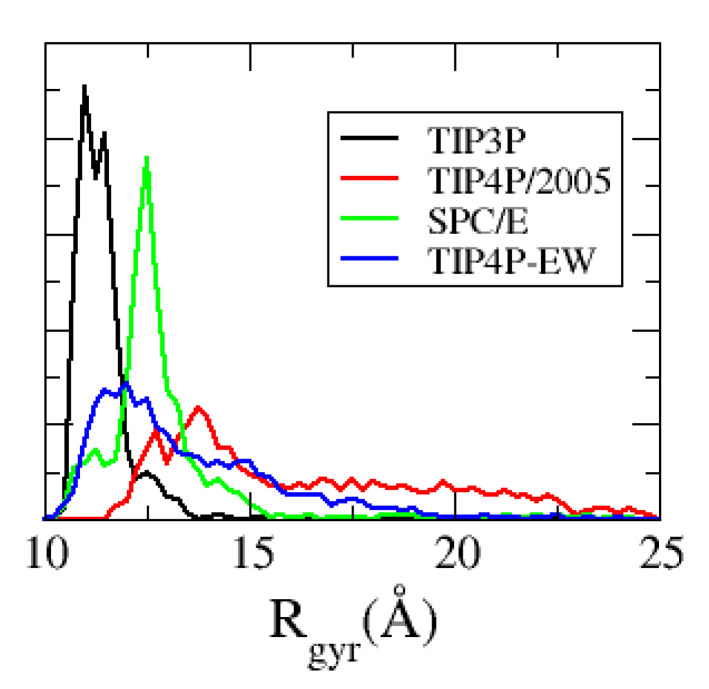
Radius of gyration distribution of the ACTR peptide, found in the simulations with the TIP3P (black line), SPC/E (green), TIP4P-EW (blue) and TIP4P/2005 (red) water models.

## Data Availability

The data presented in this study are available on request from the corresponding author.

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
