# Peer review of "Effect of the Water Model in Simulations of Protein–Protein Recognition and Association"

_polymers, 2021, doi:10.3390/polym13020176_

Round 1
Reviewer 1 Report
The paper by Agusti Emperador and coauthors describes few molecular dynamics simulations of ubiquitin and ACTR (which is an IDP) with different water models. Although I appreciate testing of different models and force fields, I suppose that results of this work are not of importance. First, such a test would be much more interesting if larger number of systems were tested. In this manuscript, only 1 well-studied protein was used, plus one figure for the 2nd protein. Second, Authors speculate about realism of water models, but according to contact map (Fig2 and 3), the simulations do not agree with NMR data: the intermolecular contact maps obtained from MD simulations and NMR experiments are completely different (for all water models). Finally, details of simulations are questionable: why Amber99SB-ILDN force field was used, although it is not the best for IDPs? Why simulations were performed in NVT? Were counter-ions used, and what was an ionic strength? It is very important for protein-protein interaction.
Fig2 - difficult to see and compare. Usage of different color for interacted residues (as at right) would be better.
Author Response
We thank the referee for his/her interesting comments. We do think that the findings of our work are relevant, since this is, as far as we know, the first time that spontaneous protein-protein dissociation has been found in simulations with a dissociation rate compatible with the experimentally measured degree of dimerization for a protein in solution (in our case ubiquitin). We consider that our work is the natural and necessary continuation to the interesting work of Abriata and Dal Peraro (ref. 2), where the authors warned about a critical shortcoming in molecular dymanics simulations: the deficiency of the generally used water models to describe the association/disssociation equilibrium in protein solutions.
We included the study of ACTR in our work because we wanted to show that TIP3P produces deficient results not only in the particular case of stable proteins, but also in the case of a disordered, extremely hydrophilic protein like ACTR. Although we find the same trend for these two extreme cases, our findings do not exclude the possibility that the description of dimerization for other proteins shows strengths and deficiencies not seen in our test cases. That, however, does not invalidate our results. On the contrary, it should stimulate similar analysis of other systems. At the same time, we want to highlight that there is little experimental data available to validate the results for other systems.
We showed less analysis for ACTR because, being a disordered protein, no structure-based analysis can be made like we did for ubiquitinin in figures 2, 3, 5 and 6. Moved by the referee's assessments we have added figure 9 depicting for ACTR the radius of gyration, which is the global structural observable that can be evaluated in the case of an IDP
We agree with the referee that figure 3 shows that the simulations with all the water models tested do not fully agree with NMR data. This a relevant fact which is the consequence of the main finding of the work, the overstabilization of nonspecific protein-protein contacts and its consequences on the associaion/dissociation balance, an important issue which has been generally overlooked so far. Figure 3 suports the main message of our work.
We chose the Amber99SB-ILDN force field because it is a highly accurate force field generally used in protein simulations, compatible with both 3-point and 4-point water models (see ref. 14), it was the most accurate force field used in the original work of Abriata and Dal Peraro (see ref. 2) and belongs to the same force field family than the force field to be used with TIP4P/2005 water, Amber03w. We did not use a force field tuned to give improved results in IDPs because our conclusions would have lost their validity if we had used different force fields for the simulations of each protein.
We made the simulations in NVT in order to control the protein concentration of the system under study. Apart from this, we used the same simulation setup as in the work by Abriata and Dal Peraro (ref. 2)
We have changed figure 2 following the referee's suggestions
Reviewer 2 Report
The manuscript by Emperador et al. reports the comparison between some water models molecular dynamics simulations, focusing on the protein-protein interaction. The manuscript also reports the comparison between the resulting models obtained with those available experimentally for two proteins representing extreme cases of protein stability and flexibility. They conclude that the water model chosen has a critical effect on protein dissociation from a complex and, therefore, these molecular dynamics simulation require further methodological improvement to be used for the prevision of protein-protein association/dissociation. The results are clearly presented and are of average interest for specialized readers.
Author Response
We kindly thank the referee for his/her revision of the manuscript
Round 2
Reviewer 1 Report
I do not change my opinion about this paper. Authors have taken into account only comments about minor changes but not about main disadvantages. I still suppose that speculations about realism of association-dissociation kinetics are meaningless if the simulated binding site is not realistic. Simulations with ACTR are also unrealistic with all water models that is obvious from Fig.9 (and even interpretation of the fig.9 is questionable – I do not see conformations with Rg near 20A in case of tip4p-ew water). However, you can easy find examples of realistic modeling of IDPs with some water models – so, the unrealistic model is not a common case. Of course, the result depends on many factors, including selecting proper parameters and force field. And one more comment about force field selection for ACTR simulations: I am afraid it is a wrong way if you select wrong FF (and the used Amber FF is not good for IDPs), get the unrealistic results and speculate about realism of water models. Of course, you probably get wrong (unrealistic) results, and even ideal water model cannot save the model. First select appropriate FF, then compare water models (and it also can be relevant since the best FF can give wrong results if the water model is bad). If you’d like to use the one FF for ordered and disordered systems, most likely the use of FF designed for IDP is the best choice.